# Physical Performance and Skeletal Muscle Transcriptional Adaptations Are Not Impacted by Exercise Training Frequency in Mice with Lower Extremity Peripheral Artery Disease

**DOI:** 10.3390/metabo13040562

**Published:** 2023-04-16

**Authors:** Jessica Lavier, Karima Bouzourène, Grégoire P. Millet, Lucia Mazzolai, Maxime Pellegrin

**Affiliations:** 1Angiology Division, Heart and Vessel Department, Lausanne University Hospital (CHUV), 1011 Lausanne, Switzerland; 2Institute of Sport Sciences, University of Lausanne, 1015 Lausanne, Switzerland

**Keywords:** exercise training frequency, lower extremity peripheral artery disease, skeletal muscle, gene expression, physical performance

## Abstract

Exercise training is an important therapeutic strategy for lower extremity peripheral artery disease (PAD). However, the effects of different exercise frequency on physiological adaptations remain unknown. Thus, this study compared the effects of a 7-week moderate-intensity aerobic training performed either three or five times/week on skeletal muscle gene expression and physical performance in mice with PAD. Hypercholesterolemic male ApoE-deficient mice were subjected to unilateral iliac artery ligation and randomly assigned to sedentary or exercise training regimens either three or five times/week. Physical performance was assessed using a treadmill test to exhaustion. Expression of genes related to glucose and lipid metabolism, mitochondrial biogenesis, muscle fiber-type, angiogenesis, and inflammation was analyzed in non-ischemic and ischemic gastrocnemius muscles by real-time polymerase chain reaction. Physical performance was improved to the same extent in both exercise groups. For gene expression patterns, no statistical differences were observed between three or five times/week exercised mice, both in the non-ischemic and ischemic muscles. Our data show that exercising three to five times a week induces similar beneficial effects on performance. Those results are associated with muscular adaptations that remain identical between the two frequencies.

## 1. Introduction

Lower extremity peripheral artery disease (PAD) is a highly prevalent public health problem affecting more than 235 million people worldwide [1]. PAD occurs when there is a significant narrowing of the arteries supplying the lower extremities, most often due to atherosclerosis. The arterial narrowing can lead to inadequate circulation and muscle ischemia during walking, inducing painful cramping (a symptom called intermittent claudication) [2]. This muscle pain in lower extremities negatively impacts the functional abilities and quality of life of patients.

According to current guidelines, exercise training (ET) is the cornerstone of conservative treatment for improving functional capacity in patients with PAD and intermittent claudication [3,4]. The clinical efficacy of ET therapy for the management of PAD and intermittent claudication is supported by many randomized controlled trials [5,6,7,8,9]. Our group has recently shown that pain-free and maximal walking distance increased, respectively, by 142% and 94% in PAD patients following 3 months of supervised ET [10]. Several potential mechanisms have been postulated to explain the benefits of ET in PAD including reduced levels of inflammatory markers, improved antioxidants response, angiogenesis, endothelium-dependent vasodilation, and skeletal muscle metabolism via an increase in oxidative enzymes [11,12]. However, the adaptive responses to ET in the lower extremity skeletal muscle enabling functional capacity improvement remain incompletely defined.

The characteristics of ET such as type or mode, frequency, intensity, time or duration, and volume can strongly affect ET-induced improvement in performance in PAD patients [12]. For example, supervised treadmill exercise is considered the most effective mode of ET for PAD patients, and the recommended duration of ET programs for optimal clinical benefits is 3 months [3,4,12]. In animal models of PAD induced by hindlimb ischemia, we and others also reported improved exercise capacity in response to ET [13,14]. In regard to the frequency, i.e., the number of exercise sessions performed per week, PAD patients are encouraged to exercise a minimum of three times per week, even though this recommendation is based on limited scientific evidence [12]. The frequency of training sessions is also an important parameter when considering the magnitude of ET-induced physiological adaptations, especially at the muscular level. Previous studies conducted by our group and others have examined the effects of different types/modes of ET (forced running, voluntary running, and forced swimming) on muscle energy metabolism and/or mitochondrial function in mouse models of PAD/hindlimb ischemia [13,15,16,17]. However, to our knowledge, no study has specifically investigated whether the frequency of ET could influence lower extremity muscle transcriptional response in PAD.

Therefore, the present study aimed to compare the effects of three versus five sessions per week of treadmill ET at moderate intensity on endurance capacity, as well as skeletal muscle expression of genes related to glucose and lipid metabolism, mitochondrial biogenesis, muscle fiber-type, angiogenesis, and inflammation in hypercholesterolemic mice with PAD. It was hypothesized that treadmill ET performed five times a week would elicit greater improvement in endurance capacity and increased expression of selected genes than the same ET performed three times a week.

## 2. Material and Methods

### 2.1. Mouse Model of Lower Extremity Peripheral Artery Disease

In this study, 8-week-old male hypercholesterolemic apolipoprotein E-deficient (ApoE^−/−^) mice on a C57BL/6J background were used. Mice were originally obtained from the Jackson Laboratory (stock number 002052) and bred at the animal facility of Epalinges (University of Lausanne). Mouse model of PAD and unilateral ischemia was generated by ligation of the right iliac artery as previously described [13]. After exposing the common iliac artery through an incision in the right lower quadrant of the abdomen, the artery was occluded with 7-0 silk suture. No surgical procedure was performed on the contralateral hindlimb. During surgery, mice were anesthetized with isoflurane (4% for induction and 1.5–2% for maintenance with 1.5 L/min oxygen) and placed on a heating pad. Chronic ischemia induction was confirmed by laser Doppler blood perfusion imaging. Mice were housed in individually ventilated cages under a 12-h light/dark cycle in a temperature- and humidity-controlled environment with ad libitum access to standard rodent chow and water. Animal procedures were approved by the cantonal committee for animal experimentation and were in accordance with the Ethical Principles and Guidelines for Experiments on Animals defined by Swiss law. All efforts were made to minimize animal suffering.

### 2.2. Exercise Training Protocols

One week after surgery, ApoE^−/−^ mice with PAD were randomly divided into three groups. Group 1: mice remained sedentary and were used as controls (SED, *n* = 6). Group 2: mice were exercised 3 times per week (Monday, Wednesday and Friday; 3X-ET, *n* = 8). Group 3: mice were exercised 5 times per week (Monday to Friday; 5X-ET, *n* = 7). ET consisted of forced running for 7 weeks on a 5-lane mouse treadmill (Panlab LE-8710, Bioseb, France). Each training session comprised 5 min warm-up at 8 cm/s and 5 min at 15 cm/s, followed by 40 min at 20 cm/s and a 5 min cool-down period at 8cm/s (0° slope). The total duration (55 min) and average intensity (approximatively 75% of maximal aerobic speed determined by the treadmill exhaustion test) of all exercise sessions were similar between the two protocols. This intensity of exercise is considered moderate according to previous studies [18,19,20]. Light electrical stimulation (0.2 mA), provided by a grid at the rear end of the treadmill belt, was used to encourage mice to run. Mice were acclimated to treadmill running for 4 consecutive days prior to surgery (10 min running at 8 cm/s).

### 2.3. Treadmill Exhaustion Test

Mice were submitted to an incremental test to exhaustion to determine endurance exercise capacity as previously described [21]. After a 5 min warm-up at 8 cm/s, the running speed was increased by 2 cm/s every 3 min until exhaustion (0° slope). Exhaustion was defined when mice stayed on the electrical grid at the rear of the treadmill for 5 consecutive sec without attempting to reengage the treadmill belt. The test was performed at baseline (i.e., one week after surgery) and at the end of the study (i.e., 8 weeks after surgery). The maximal running distance (m), time (min), and speed (cm/s) were determined for each mouse.

### 2.4. Gene Expression Analysis by Real-Time Polymerase Chain Reaction

About 48 h after the last training session, right ischemic and left contralateral non-ischemic gastrocnemius muscles were collected, snap frozen in liquid nitrogen, and stored at −80 °C until use. Total RNA was extracted using the RNeasy Fibrous Tissue Mini Kit (Qiagen, Hombrechtikon, Switzerland) according to the manufacturer’s protocol. In brief, ≈30 mg of frozen tissue samples was immersed in lysis buffer and disrupted for 2 × 3 min at 30 Hz using the Tissue Lyser II and stainless steel beads (Qiagen). Samples were digested with Proteinase K prior to extraction and a DNase digestion step was included in the protocol. RNA quantity and purity (absorbance ratio at 260/280) were determined using a spectrophotometer (Nanodrop 2000C, Thermo Fisher Scientific, Waltham, MA, USA). All RNA samples had a 260/280 ratio > 2. One μg of RNA was then reverse transcribed into cDNA using the PrimeScript RT Reagent Kit with gDNA eraser (TaKaRa Bio Inc., Shiga, Japan) according to the manufacturer’s protocols. Briefly, genomic DNA was removed by mixing 2 µL 5× gDNA Eraser Buffer and 1 µL gDNA Eraser with 7 μL template RNA followed by incubation in CFX96 Touch Real-Time PCR Detection System (Bio-Rad Laboratories Inc, Cressier, Switzerland) for 2 min at 42 °C before a hold at 4 °C. Subsequently, 10 μL of reverse-transcription master mix comprising 4 µL of 5× PrimeScript Buffer, 1 µL of PrimeScript RT Enzyme Mix I, 1µL of RT primer Mix and 4 µL of RNase free water were added to 10 µL of each sample. The final reaction was incubated in the thermal cycler at 37 °C for 15 min, followed by incubation at 85 °C for 5 sec before a hold at 4 °C. For quantitative PCR, 5 μL of TB Green Premix Ex Taq (TaKaRa Bio Inc., Japan), 0.5 μL of 5 μM forward and reverse primer, 1.5 μL of nuclease free water, and 2.5 μL of cDNA were mixed to a total volume of 10 μL. The PCR reactions were run in duplicate on a CFX96 Touch Real-Time PCR Detection System, under the following conditions: initial denaturation at 95 °C for 30 sec, followed by 40 cycles at 95 °C for 5 s and 60 °C for 30 s. The target genes evaluated were: solute carrier family 2 member 1 (GLUT1), solute carrier family 2 member 4 (GLUT4), hexokinase 2 (HK2), phosphofructokinase (PFK), pyruvate dehydrogenase kinase 4 (PDK4), glycogen synthase 1 (GYS1), lactate dehydrogenase A (LDHA), lactate dehydrogenase B (LDHB), monocarboxylate transporter 1 (MCT1), monocarboxylate transporter 4 (MCT4), fatty acid translocase cluster of differentiation (CD36), fatty acid binding protein 3 (FABP3), carnitine palmitoyltransferase 1beta (CPT1β), long-chain acyl-CoA dehydrogenase (LCAD), mitochondrial uncoupling protein 2 (UCP2), hormone-sensitive lipase (HSL), peroxisome proliferator-activated receptor gamma (PPAR-γ), peroxisome proliferator-activated receptor gamma coactivator 1-alpha (PGC-1α), peroxisome proliferator-activated receptor gamma coactivator 1-beta (PGC-1β), nuclear respiratory factor 1 (NRF1), Mitochondrial transcription factor A (TFAM), myosin heavy chain 7 (MyH7), myosin heavy chain 2 (MyH2), myosin heavy chain 1 (MyH1), myosin heavy chain 4 (MyH4), vascular endothelial growth factor A (VEGFA), hypoxia-inducible factor 1 alpha (HIF-1α), interleukin-1 beta (IL-1β), cluster of differentiation 11c (CD11c), interleukin-10 (IL-10), and interleukin-1 receptor antagonist (IL-1ra). The primer sequences used are listed in Appendix A and were purchased from Microsynth (Balgach, Switzerland). Gene expression was normalized using 36B4 as a housekeeping gene. The 2^−ΔΔCt^ method was used to calculate the fold change expression levels in exercised groups over sedentary group.

### 2.5. Statistical Analysis

Statistical analyses were conducted with Prism 9.1.0 for Windows (GraphPad, La Jolla, CA, USA). All data are presented as mean ± SEM. Data were analyzed using two-way repeated measures ANOVA or one-way ANOVA followed by Holm–Sidak’s multiple comparisons tests as stated in the Figure legends. A value of *p* < 0.05 was considered to be statistically significant.

## 3. Results

### 3.1. Physical Performance

Figure 1 shows maximal running distance (MRD), maximal running time (MRT), and maximal running speed (MRS) at baseline and the end of the study for each group. There were no significant differences in baseline MRD, MRT, or MRS between the three groups. MRD, MRT, and MRS were improved in 3X-ET mice (492 ± 170 vs. 312 ± 67 m, Figure 1A; 42.7 ± 7.7 vs. 33.0 ± 4.5 min, Figure 1B; 34.3 ± 5.4 vs. 27.8 ± 2.7 cm/s, Figure 1C, respectively) (*p* < 0.05). All parameters were also improved in 5X-ET mice between end of the study and baseline (MRD: 539 ± 296 vs. 260 ± 106 m, Figure 1A; MRT: 44.0 ± 14.2 vs. 28.9 ± 7.6 min, Figure 2B; MRS: 35.4 ± 9.6 vs. 24.9 ± 5.5 cm/s, Figure 3C) (*p* < 0.01). At end of the study, neither MRD, MRT, nor MRS differed between 3X-ET and 5X-ET mice (Figure 1). No improvement was observed in SED mice for any of these parameters (Figure 1).

### 3.2. Gene Expression Related to Glucose and Lactate Metabolism

GLUT1, GLUT4, GYS1, HK2, PFK, PDK4, LDHA, LDHB, MCT1, and MCT4 mRNA levels were analyzed in non-ischemic and ischemic muscles for the three groups. As shown in Figure 2A, mRNA expression of GLUT1 was decreased in the non-ischemic muscle of 3X-ET and 5X-ET compared to the non-ischemic muscle of SED (0.57-fold change and 0.56-fold change vs. SED, respectively, *p* < 0.05). GLUT1 mRNA expression was also decreased in the ischemic muscle of 3X-ET and 5X-ET in comparison with the ischemic muscle of SED (0.52-fold change and 0.54-fold change vs. SED, respectively, *p* < 0.05) (Figure 2A). GLUT4 mRNA expression was significantly decreased in 3X-ET (0.58-fold change vs. SED, *p* < 0.05) and tended to decrease in 5X-ET (0.64-fold change, *p* = 0.08 vs. SED) in the ischemic muscle, while no significant difference was observed between ET groups and SED in the non-ischemic muscle (Figure 2A). GYS1 mRNA expression did not differ between ET groups and SED, either in the non-ischemic or in the ischemic muscle (Figure 2B). HK2 mRNA expression was decreased in 3X-ET and 5X-ET in the ischemic muscle (0.47-fold change, *p* < 0.01 and 0.52-fold change vs. SED, respectively, *p* < 0.05), but not in the non-ischemic one (Figure 2C). Compared to SED, mRNA expression of PFK decreased in the non-ischemic muscle both in 3X-ET and 5X-ET (0.46-fold and 0.33-fold change vs. SED, respectively, *p* < 0.01) (Figure 2C). In the ischemic muscle, PFK expression was also decreased significantly in 3X-ET (0.57-fold change vs. SED, *p* < 0.05) and tended to decrease in 5X-ET (0.65-fold change vs. SED, *p* = 0.06) (Figure 2C). There is no statistically significant difference in PDK4 mRNA expression between the groups (Figure 2C).

Figure 3A showed that LDHA mRNA expression was only decreased in the ischemic muscle in 3X-ET and 5X-ET (0.62-fold and 0.68-fold change vs. SED, respectively, *p* < 0.05). Expression of LDHB decreased both in the non-ischemic (0.51-fold in 3X-ET and 0.44-fold change in 5X-ET vs. SED, *p* < 0.05) and non-ischemic muscles (0.48-fold in 3X-ET and 0.44-fold change in 5X-ET vs. SED, *p* < 0.01). MCT1 mRNA expression decreased in the ischemic muscle both in 3X-ET and 5X-ET (0.46-fold and 0.33-fold change vs. SED, respectively, *p* < 0.05) (Figure 3B). In the non-ischemic muscle, there was a significant decrease in MCT1 expression in 5X-ET (0.41-fold change vs. SED, *p* < 0.05), but not in 3X-ET (*p* = 0.15) (Figure 3B). MCT4 mRNA level in the non-ischemic muscle of the ET groups was not different to those of the SED group (Figure 2E). Compared to SED, the level of MCT4 mRNA showed a trend toward a decrease in the ischemic muscle of 3X-ET (0.59-fold change vs. SED, *p* = 0.06) and 5X-ET (0.65-fold change vs. SED, *p* = 0.10) (Figure 3B).

The comparison between 3X-ET and 5X-ET revealed no significant difference in mRNA expression for any of these genes (Figure 3).

### 3.3. Gene Expression Related to Lipid Metabolism

As shown in Figure 4A, neither CD36 nor FABP3 mRNA levels were different between 3X-ET and SED as well as between 5X-ET and SED, either in the non-ischemic or in the ischemic muscle. Compared to SED, CPT1β and LCAD mRNA levels were decreased both in the non-ischemic (0.46-fold and 0.55-fold change vs. SED, respectively, *p* < 0.01) and ischemic (0.55-fold and 0.57-fold change, respectively, *p* < 0.05) muscles of 3X-ET (Figure 4B). The mRNA expression of these two genes was also decreased in the two muscles of 5X-ET (For CPT1β: 0.33-fold change vs. SED in the non-ischemic muscle, *p* < 0.01 and 0.58-fold change vs. SED in the ischemic muscle, *p* = 0.06; for LCAD: 0.58-fold change vs. SED in the non-ischemic muscle, *p* < 0.05 and 0.64-fold change vs. SED in the ischemic muscle, *p* < 0.05) (Figure 4B). Regarding UCP2, HSL, and PPAR-γ, mRNA expression was unchanged between ET groups and SED, either in the non-ischemic or in the ischemic muscle (Figure 4B).

No significant differences were found in mRNA expression between 3X-ET and 5X-ET for any of these genes (Figure 4).

### 3.4. Gene Expression Related to Mitochondrial Biogenesis

As shown in Figure 5, neither PGC1-α, PGC1-β, NRF1 nor TFAM mRNA levels were different between groups for any muscle.

### 3.5. Gene Expression Related to Muscle Fiber-Type and Angiogenesis

Expression of genes encoding MyH7 (marker of slow-twitch type 1 fibers), MyH2 (marker of fast-twitch type IIa fibers), MyH1 (marker of fast-twitch type IIx fibers), and MyH4 (marker of fast-twitch type IIb fibers) are shown in Figure 6A. Our results revealed a lower MyH7 mRNA expression in the ischemic muscle of 5X-ET compared to that of the SED (0.29-fold change vs. SED, *p* < 0.05). No significant difference in mRNA expression for any of these four genes was observed between 3X-ET and 5X-ET (Figure 6A).

As shown in Figure 6B, there was not any significant difference between ET groups and SED, as well as between the two ET groups in VEGFA and HIF-1α mRNA levels.

### 3.6. Gene Expression Related to Inflammation

Figure 7 shows mRNA expression of pro-inflammatory (IL-1β and CD11c) and anti-inflammatory (IL-10 and IL-1ra) markers. Neither in the non-ischemic muscle nor in the ischemic muscle, IL-1β, CD11c, IL-10, or IL-1ra mRNA expression differed between the three groups.

## 4. Discussion

In the present study, we investigated the effects of a 7-week moderate-intensity running training program with different frequencies (three vs. five times a week) on physical performance and gene expression patterns in ischemic and non-ischemic lower limb skeletal muscle.

The main findings are as follows: (i) at the end of the exercise program, both exercise groups showed significant increase in physical performance compared with baseline (pre exercise program), with no significant difference between the two exercise frequencies; (ii) expression of genes involved in energy metabolism was moderately modulated by exercise in non-ischemic and ischemic lower limb muscle at the end of the study, but with no significant effect of exercise frequency; (iii) exercise did not remarkably change muscular expression of genes related to mitochondrial biogenesis, muscle fiber-type, or inflammation, regardless of the frequency.

For PAD patients, a typical exercise program consists of supervised treadmill exercise at least three times per week for a minimum of 12 weeks [12,22]. Each session includes a minimum of 30 min of exercise and requires patients to walk to maximum ischemic pain [12,22]. However, the optimal characteristics of ET, in particular the frequency, for improving the functional capacity in PAD patients are not well established. In a study from Nakamura et al., it was reported that older adult women enrolled in a 12-week exercise program improved their 6 min walking distance (i.e., a widely used test to assess aerobic capacity or endurance) more with a frequency of three times compared to once or twice a week [23]. On the other hand, there was no difference in functional performance in older adults following an 8-week resistance training performed either two or three days/week [24]. In patients with PAD and intermittent claudication, Nicolaï et al. showed that the clinical benefit in walking performance was similar between PAD patients who exercised once, twice, or three times a week for 12 weeks [25]. We showed here for the first time that ET three and five times a week was equally effective in improving endurance capacity in a mouse model of PAD. In line with our preclinical study, Le Bris et al. demonstrated that three or five days of training per week induced similar increases in endurance capacity in coronary artery disease patients [26]. Our results indicate that weekly frequency may not be the most important characteristic for optimizing the benefits of ET on functional capacity, in the setting of PAD. Although it is considered a first-line treatment for symptomatic PAD patients, adherence to supervised exercise therapy is poor [27]. From a clinical point of view, it is thus of interest that low-frequency ET provides the same benefits as high-frequency ET. In fact, a lower-frequency ET may enable a greater number of PAD patients to engage in ET and lead to higher adherence. Further clinical investigations are necessary to better understand the role of the frequency of ET on functional capacity as well as quality of life in PAD patients.

There is a limited number of studies focusing on the individual role of frequency of aerobic ET on skeletal muscle adaptations. For example, Silva et al. reported that 8 weeks of treadmill running performed five times per week was more effective than the same training performed three times per week in increasing the mitochondrial respiratory chain enzyme activities as well as decreasing the oxidative stress in the quadriceps muscle in mice [28]. To the best of our knowledge, the relationship between the frequency of exercise and the resulting adaptations in lower limb skeletal muscle has not been previously investigated in PAD.

In the present study, we first focused on mRNA expression of genes implicated in glucose metabolism pathways. In the non-ischemic muscle, our data indicated a similar downregulation of mRNA expression of GLUT1 (a predominant glucose transporter in skeletal muscle alongside GLUT4) and PFK (a key glycolysis rate-limiting enzyme alongside HK2) in three and five exercise sessions per week. Interestingly, the effect of exercise training on glucose metabolism genes was slightly more pronounced in the ischemic muscle than in the non-ischemic one as evidenced by a downregulation of mRNA levels of GLUT1 and PFK (as it occurred in the non-ischemic muscle) but also of HK2 and GLUT4. These results in the five times/week exercised group are inconsistent with our previous findings in the same mouse model showing no effect of five times a week treadmill running on the mRNA expression of these four genes in the ischemic gastrocnemius muscle [13]. Differences in the duration of the treadmill running program (7 weeks in the present study vs. 4 weeks in our previous study) and/or protocol (continuous running at 20 cm/s for 40 min in the present study vs. 32 cm/s until exhaustion in our previous study) might explain the discrepancies between the results. It is of importance to note that we did not detect any significant changes between the two exercise frequencies in the expression of the 10 analyzed genes participating in glucose metabolism. This lack of difference might explain why we did not observe any difference in endurance capacity improvement between the two ET frequencies groups.

To our knowledge, the present study was the first to investigate the expression of genes regulating lactate metabolism in response to ET in the setting of PAD. During exercise, lactate is produced during glycolysis by the degradation of glucose-6-phosphate in the contracting muscle. It is now recognized that lactate is an important energy substrate for skeletal muscle fibers during exercise, as well as a key factor for exercise-induced mitochondrial adaptations [29]. In the non-ischemic hindlimb muscle, our data revealed that three and five times per week ET significantly downregulated LDHB mRNA level to the same degree, whilst no significant downregulation was observed for LDHA. On the contrary, Gill et al. demonstrated a significantly reduced LDHA but not LDHB mRNA levels in non-ischemic quadriceps muscle of C57BL/6 mice exercised three times per week on a treadmill for 12 weeks [30]. Another recent study showed increased LDHB mRNA expression after 6 weeks of voluntary wheel running in mouse quadriceps muscle [31]. These differences might be explained by the discrepancies in the training program durations/protocols and/or the analyzed muscles between the studies. Regarding the latter point, it has been demonstrated that the expression patterns of LDHB are different according to the muscle fiber type in mice [32]. In human vastus lateralis muscle samples, an elevated expression of LDHB but not LDHA was reported following a 3-week exercise program performed four times a week [32]. Clearly, more studies are needed to better understand the changes in LDH isoforms gene expression with ET. In the ischemic muscle, however, we demonstrated a similar downregulation of both LDHA and LDHB mRNA levels following three times and five times per week ET. Because LDHA favors the reaction that converts pyruvate to lactate, whereas LDHB favors the reverse reaction producing pyruvate from lactate, we can speculate that ET, independently of the frequency, has no effect on lactate/pyruvate production in the ischemic muscle in our mouse model. To further explore muscle lactate metabolism, we looked at the expression of the main lactate transporters MCT1 and MCT4. As part of the intracellular lactate shuttle, MCT1 facilitates the uptake of lactate molecules from the circulation into skeletal muscle cells for oxidative metabolism, while MCT4 facilitates lactate extrusion out of muscle [33]. In the non-ischemic muscle, we reported no significant effect of three times per week exercise on MCT1 and MCT4 mRNA expression, whilst five times per week exercise significantly downregulated MCT1 expression only. Ahmadi et al. recently reported in lower limb muscles of rats that MCT1 and MCT4 mRNA levels did not decrease following an 8-week treadmill running protocol performed at a frequency of five times per week [34]. Interestingly, we observed an almost similar decrease (although not significant) in MCT1 and MCT4 mRNA levels in the ischemic muscle following the two training frequencies. Thus, it could be hypothesized that exercise frequency did not affect lactate transport into the ischemic muscle cells.

Next, we examined specific changes in expression of genes involved in fatty acid oxidation pathway. Among the seven genes investigated (CD36, FABP3, LCAD, CPT1β, UCP2, HSL, and PPAR-γ), exercise training only decreased the mRNA levels of CPT1β (catalyzing the first step in long-chain fatty acid import into mitochondria) and LCAD (catalyzing the initial step in mitochondrial fatty acid oxidation) in both the non-ischemic and ischemic muscles. Once again, no significant difference between the two exercise frequencies was observed. In a previous study, we demonstrated an increased mRNA expression of CD36, FABP3, CPT1β, HSL, and UCP2 in the non-ischemic gastrocnemius muscle of PAD mice that underwent a 4-week treadmill program with five sessions per week [13]. In the same study, we saw an increase in FABP3 and CPT1β mRNA expression in the ischemic muscle. Again, as outlined above, the differences in the exercise protocols between our two studies (i.e., continuous treadmill running for 7 weeks at 20 cm/s for 40 min vs. 4 weeks at 32 cm/s until exhaustion our previous study, respectively) might be a reason for this discrepancy. In line with the results of glucose metabolism, this similar effect of both exercise frequencies on the expression of genes involved in lipid metabolism is consistent with the data on endurance exercise capacity (i.e., similar improvement with the two ET frequencies).

Another interesting finding of this study is that the mRNA expression of transcription factors for mitochondrial biogenesis genes (PGC-1α, PGC-1β, NRF1 and TFAM) was unaltered regardless of exercise training frequency in our mouse model. Previous reports from our group and others also showed no effect of treadmill running five [13,34] or three times per week [35,36] on PGC-1α expression at the transcriptional level in non-ischemic hindlimb muscle of rodents. In the ischemic muscle, we also reported no obvious effect of five times a week treadmill running on PGC-1α, PGC-1β, NRF1, and TFAM [13]. In line with our study, Nagase et al. demonstrated no increase in ischemic muscle PGC-1α gene expression after three weeks of treadmill exercise performed twice a week despite increased endurance performance [15]. In contrast, Albadawil et al. showed enhanced PGC-1α mRNA expression in ischemic muscle of mice that underwent 60 min of daily treadmill exercise over 4 weeks [17]. Additional research is to be considered to clarify the role of exercise frequency on mitochondrial biogenesis in skeletal muscle ischemic disorders.

Regarding mRNA levels of MyHC isoforms, we observed a significant reduction in MyH7 in the ischemic muscle in response to five times/week exercise compared to non-exercised animals. However, we could not detect any significant differences between the two exercise frequencies either in the non-ischemic or ischemic muscle. These results are not surprising and are in line with those related to muscle metabolism and mitochondrial biogenesis gene expression for which we reported no effect of exercise frequency as well. Earlier reports also showed no significant effect of ET at various frequencies on expression of genes encoding MyHC in gastrocnemius muscle of mice (without PAD) [36,37].

In this work, we showed that neither the expression of VEGFA nor that of HIF-1α were different between PAD mice exercised three or five times/week. Interestingly, the effect of exercise frequency on mouse gastrocnemius (non-ischemic) muscle VEGFA mRNA level has been previously investigated [38]. In agreement with our data, Kivela et al. reported no difference in muscle VEGFA mRNA expression between trained and sedentary mice, but also between three and five times per week exercised mice [38]. In ischemic hindlimb muscle tissues, previous studies from our lab and from Albadawil et al. also showed no effect of treadmill running five times/week [13] and seven times/week [17] on VEGFA and/or HIF-1α mRNA expression. Altogether, one could hypothesize that exercise frequency does not modulate VEGFA and/or HIF-1α expression in ischemic and non-ischemic muscles, at least at the transcriptional level.

There is growing interest in the role of ET on systemic and muscular inflammation in PAD. In gastrocnemius tissue samples of PAD patients, Andrade-Lima et al. recently reported that a 12-week submaximal walking training (two sessions/week) decreased mRNA levels of pro-inflammatory markers IL-6, TNF-α, and C reactive protein [39]. In this work, we did not report any effect of our ET protocols on mRNA expression of pro-inflammatory cytokine IL-1β and macrophage marker CD11c as well as anti-inflammatory cytokines IL-10 and IL-1ra. It would be interesting to further investigate how exercise as a whole, and more specifically exercise frequency, impacts inflammatory response in PAD.

In the present study, the duration of exercise sessions (55 min), but not the training load, was similar between the three and five times a week exercise protocols. However, it is interesting to note that even if the training load was higher in five times a week exercised mice (1925 min) compared to three times a week exercised ones (1125 min), physical performance and gene expressions patterns were almost similar between the two groups. This indicates that training load might not be the primary factor that influences physiological and transcriptional adaptations in our mouse model. Although this study focused on ET frequency, we cannot rule out that exercise intensity, independent of the training load, is paramount, and modulates adaptations in skeletal muscles but also in many other organs [40,41,42].

## 5. Conclusions

In summary, the present study demonstrated that physical performance as well as gene expression related to glucose and fatty acid metabolism, mitochondrial biogenesis, muscle fiber-type, angiogenesis, and inflammation, were not modulated by the frequency of moderate-intensity aerobic training in a mouse model of PAD. Exercise frequency—at least three and five times per week—appears to be a minor characteristic mediating the amplitude of muscle transcriptional adaptations and physical performance in response to exercise in PAD. Further clinical investigations are necessary to determine whether our findings are confirmed in PAD patients.

## Figures and Tables

**Figure 1 metabolites-13-00562-f001:**
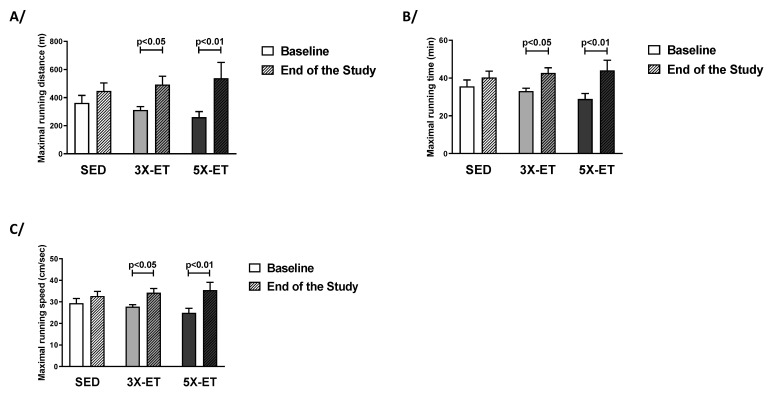
Exercise endurance capacity in sedentary (SED), three times (3X-ET), and five times (5X-ET) a week exercised ApoE^−/−^ mice with PAD. Maximal running distance (**A**), maximal running time (**B**), and maximal running speed (**C**) were determined by an incremental treadmill running test to exhaustion (*n* = 6–8 mice per group). Results presented as the mean ± SEM. Data were analyzed using two-way repeated measures ANOVA with Holm–Sidak’s post hoc tests.

**Figure 2 metabolites-13-00562-f002:**
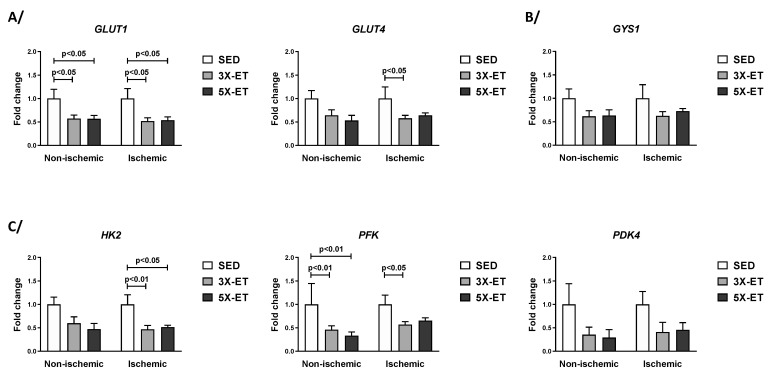
Expression of genes involved in glucose transport (*GLUT1* and *GLUT4*) (**A**), glycogen synthesis (*GYS1*), (**B**) and glycolysis (*HK2*, *PFK,* and *PDK4*) (**C**) in non-ischemic and ischemic muscles of sedentary (SED), three times (3X-ET), and five times (5X-ET) a week exercised ApoE^−/−^ mice with PAD. Data were normalized to 36B4 gene and expressed as fold-change over respective SED group. Results presented as the mean ± SEM (*n* = 6–8 mice per group). Data were analyzed using one-way ANOVA with Holm–Sidak’s post hoc tests.

**Figure 3 metabolites-13-00562-f003:**
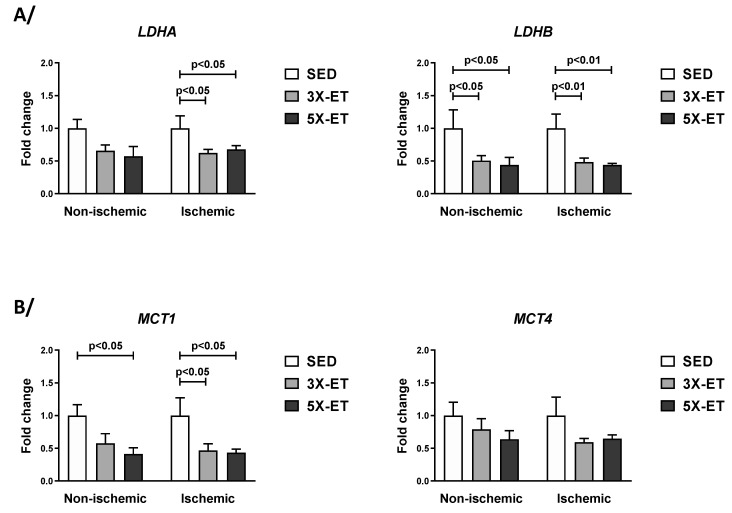
Expression of genes involved in lactate production (*LDHA* and *LDHB*) (**A**) and lactate transport (*MCT1* and *MCT4*) (**B**) in non-ischemic and ischemic muscles of sedentary (SED), three times (3X-ET), and five times (5X-ET) a week exercised ApoE^−/−^ mice with PAD. Data were normalized to 36B4 gene and expressed as fold-change over respective SED group. Results presented as the mean ± SEM (*n* = 6–8 mice per group). Data were analyzed using one-way ANOVA with Holm–Sidak’s post hoc tests.

**Figure 4 metabolites-13-00562-f004:**
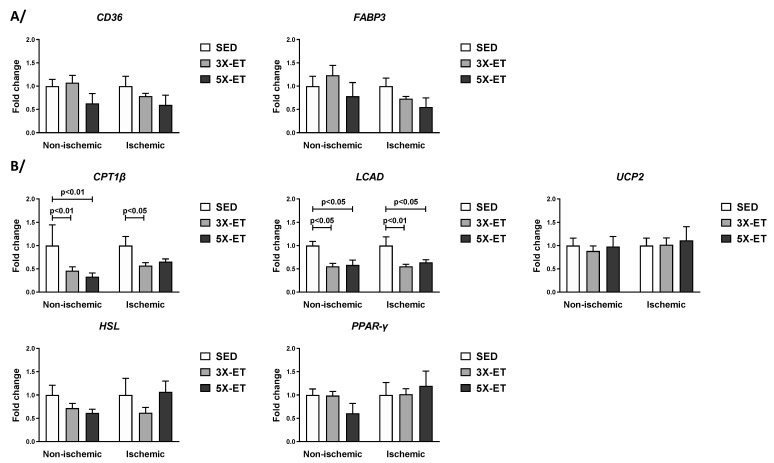
Expression of genes involved in fatty acid uptake (*CD36*) and transport (*FABP3*) (**A**), and fatty acid β-oxidation (*CPT1β*, *LCAD*, *UCP2*, *HSL* and *PPAR-γ*) (**B**) in non-ischemic and ischemic muscles of sedentary (SED), three times (3X-ET), and five times (5X-ET) a week exercised ApoE^−/−^ mice with PAD. Data were normalized to 36B4 gene and expressed as fold-change over respective SED group. Results presented as the mean ± SEM (*n* = 6–8 mice per group). Data were analyzed using one-way ANOVA with Holm–Sidak’s post hoc tests.

**Figure 5 metabolites-13-00562-f005:**
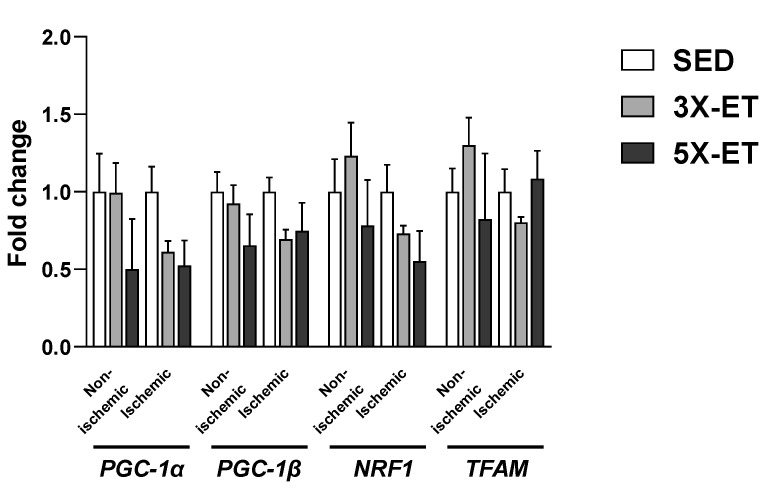
Expression of genes involved in mitochondrial biogenesis (*PGC-1α*, *PGC-1β*, *NRF1* and *TFAM*) in non-ischemic and ischemic muscles of sedentary (SED), three times (3X-ET), and five times (5X-ET) a week exercised ApoE^−/−^ mice with PAD. Data were normalized to 36B4 gene and expressed as fold-change over respective SED group. Results presented as the mean ± SEM (*n* = 6–8 mice per group). Data were analyzed using one-way ANOVA with Holm–Sidak’s post hoc tests.

**Figure 6 metabolites-13-00562-f006:**
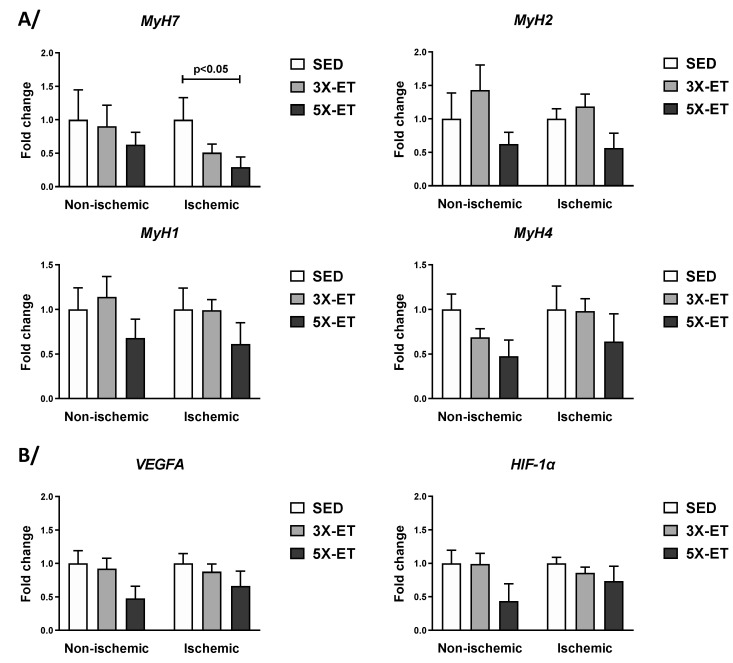
Expression of genes coding for MyHC isoforms (*MyH7*, *MyH2*, *MyH1* and *MyH4*) (**A**) and related to angiogenesis (*VEGFA* and *HIF-1α*) (**B**) in non-ischemic and ischemic muscles of sedentary (SED), three times (3X-ET), and five times (5X-ET) a week exercised ApoE^−/−^ mice with PAD. Data were normalized to 36B4 gene and expressed as fold-change over respective SED group. Results presented as the mean ± SEM (*n* = 6–8 mice per group). Data were analyzed using one-way ANOVA with Holm–Sidak’s post hoc tests.

**Figure 7 metabolites-13-00562-f007:**
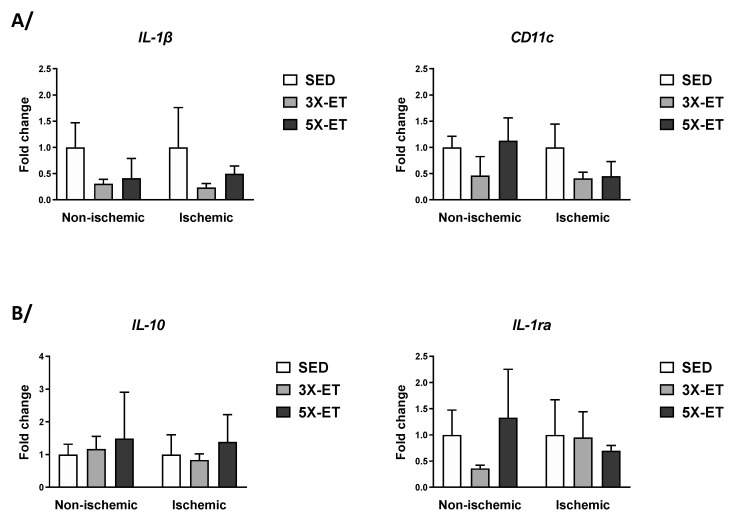
Expression of genes related to pro-inflammatory markers (*IL-1β* and *CD11c*) (**A**) and anti-inflammatory cytokines (*IL-10* and *IL-1ra*) (**B**) in non-ischemic and ischemic muscles of sedentary (SED), three times (3X-ET), and five times (5X-ET) a week exercised ApoE^−/−^ mice with PAD. Data were normalized to 36B4 gene and expressed as fold-change over respective SED group. Results presented as the mean ± SEM (*n* = 4–5 mice per group). Data were analyzed using one-way ANOVA with Holm–Sidak’s post hoc tests.

## Data Availability

The original data of this study are available from the corresponding author upon reasonable request. Data is not publicly available due to privacy or ethical restrictions.

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
