# Peer review of "Physical Performance and Skeletal Muscle Transcriptional Adaptations Are Not Impacted by Exercise Training Frequency in Mice with Lower Extremity Peripheral Artery Disease"

_metabolites, 2023, doi:10.3390/metabo13040562_

Round 1

Reviewer 1 Report

I read with great pleasure the paper by Lavier and colleagues regarding the impact of exercise training frequency on physical performance and skeletal muscle transcriptional in a mouse model of lower extremity peripheral artery disease.

Overall, the paper is interesting and well-written. However, when dealing with exercise training, one cannot merely focus on the frequency, without taking into consideration other key parameters such as volume (intensity) and duration, as correctly observed by the Authors in the Introduction (see lines “The characteristics of ET such as type or mode, frequency, intensity, time or duration, and volume can strongly affect ET-induced improvement in performance in PAD patients”). Furthermore, in this part of the paper, reference is made to study performed on patients; however, it would be more appropriate if the Authors refer also to animal studies reported in the literature and compare the type of exercise protocol with others already published in the literature. In the reviewer’s opinion, a critical point is the definition of the level (intensity) of treadmill training (low, moderate, or high treadmill training). In fact, as reported in the current literature, distinct exercise training schedules of different intensities can cause a number of physiological adaptations in different organs, and especially at the level of the adrenal gland, which represents an early target of PE. In particular, evidence shows that the hypothalamus-pituitary-adrenocortical axis, as well as the sympatho-adrenomedullary system, is mainly involved in mediating the physiological effects of PE. Again, several morphological and biochemical changes were reported with reference to oxidative metabolism and muscle fiber composition in the mouse. In line with this, while slight to moderate levels seemed to be beneficial to cartilage health, more strenuous exercise may be detrimental. Thus, if exercise is thought to act like as a disease-modifying agent, the dosage may be critical to its success. However, from the paper, it does not clearly emerge the characteristic and intensity of treadmill PE. Therefore, in the opinion of the reviewer, it is important that the Author implement the introduction by discussing and citing previous studies, some key of them reported below, and clearly discuss all these points and compare their PE treadmill protocol with previous literature to highlight whether it refers to a low, moderate or high-intensity treadmill training. This also applies for the paragraph 2.2 of materials and methods section where the Authors should specify according to these articles what level of intensity is obtained in their protocol (see lines “The total duration (55 min) and average intensity (about 75% of the maximum aerobic speed determined by the treadmill exhaustion test) of all exercise sessions exercise were similar between the two protocols”). Such a revision of the paper should not be intended in making novel experiments by varying more parameters, but at least in discussing this aspect citing relevant literature in various sections (e.g. intro, methods, discussion), since this would add significance to the paper.

Bartalucci et al.Histol Histopathol. 2012 Jun;27(6):753-69. doi: 10.14670/HH-27.753. PMID: 22473696.

Toti et al. Biol Sport. 2013 Dec;30(4):301-9. doi: 10.5604/20831862.1077557. Epub 2013 Nov 25. PMID: 24744502; PMCID: PMC3944543

Shimomura et al. Int J Mol Sci. 2018 Jun 3;19(6):1653. doi: 10.3390/ijms19061653. PMID: 29865282; PMCID: PMC6032207.

Zhou et al. Biomed Eng Online. 2021 Nov 18;20(1):111. doi: 10.1186/s12938-021-00949-6. PMID: 34794451; PMCID: PMC8600697

Another major point is that Figure 5 is missing in the text. 

Reviewer 2 Report

The present study by Lavier et al. investigates the effects of running training frequencies (three vs. five times a week) on physical performance and gene expression patterns in ischemic and non-ischemic lower limb skeletal muscle. The study's main findings suggest that both exercise groups showed a significant increase in physical performance compared to the baseline, with no significant difference between the two exercise frequencies. Furthermore, exercise did not significantly alter the muscular expression of genes related to mitochondrial biogenesis, muscle fiber-type, or inflammation, regardless of frequency. The authors suggest that low-frequency exercise provides the same benefits as high-frequency exercise. Overall, the study provides valuable insights into the effects of exercise frequency on physical performance and gene expression in skeletal muscle; however, further data is necessary to support the conclusions.

Major comments:

1.      The authors should emphasize their study's significance as it is unclear in the article. They mention a similar study conducted by Pellegrin et al. (2020) with a different protocol, but it is unclear how their study adds to the existing literature.

2.      The rationale for performing relative gene expression analysis of lactate metabolism genes should be provided in the discussion.

3.      The authors mention that their findings are inconsistent with their previous findings and other reports with respect to LDHB, but they do not provide a detailed discussion on why this might be the case, and the authors only mention protocol differences. Further discussion is needed.

4.      The authors should provide evidence to support the claim that glucose metabolism is more pronounced than lactate metabolism.

5.      To further support their claim of no significant change in myofiber type, the authors can perform muscle myosin stains on the fibers.

6.      Figure 5 graphs are missing in the current version of the manuscript.

7.      The methods need to clarify the exhaustion time for the mice.

Minor comments:

All figures can be reorganized for better readability. For example, in Figure 4, all the graphs can be merged into one.

The authors should consider following a consistent theme in arranging the graphs. For instance, Figure 2 can be split into two figures: glucose metabolism and lactate metabolism.

Reviewer 3 Report

The manuscript of Lavier et al. entitled: ”Physical performance and skeletal muscle transcriptional adaptations are not impacted by exercise training frequency in mice with lower extremity peripheral artery disease” tested the hypothesis that more frequent exercise training (3 versus 5 times/week) induces greater improvement in physical performance in mice with lower extremity peripheral artery disease (PAD). Testing endurance capacity and gene expression related to glucose and lactate metabolism, lipid metabolism, mitochondrial biogenesis, and muscle fiber-type and angiogenesis, they demonstrated that skeletal muscle adaptation is not dependent on the frequency of aerobic training in a mouse model of PAD. Reading the manuscript and appreciating the arguments, I identified the following major and minor issues which need to be addressed by authors.

Major concerns:

1. Hypercholesterolemic apolipoprotein E-deficient (ApoE-/-) mice already develop atherosclerotic lesions in adulthood. Therefore, it is not clear to me, why ligation of the right iliac artery had to be performed to induce animal model of PAD? The authors should clearly explain their strategy.

2. Figure 1: It would be appropriate to also show statistical comparison of SED with 3x-ET and 5x-ET groups to demonstrate reliability of a PAD model.

3. Figure 2: Despite mRNA level of all tested proteins were decreased to approximately 50%, statistical significance in respect to SED was not found. It is quite surprising. It seems to me that value of P was not very far from 0.05. Thus, adding more data might reveal significance. The authors should consider this aspect. In addition, it is known that exercise training is the potent stimulus to increase skeletal muscle GLUT4 expression. However, a decreasing trend is presented in this work. This does not corelate well with the beneficial effect of training showed in Figure 1. Because gene expression related to lipid metabolism, mitochondrial biogenesis, and muscle fiber-type and angiogenesis was not changed a lot by training, it appears that only compromised glucose and lactose metabolism could be implicated in better physical performance of mice with PAD. It, however, contrasts sharply with transcriptional adaptation of healthy skeletal muscle to physical training. The authors should discuss this aspect.

4. Figure 5 is missing.

5. The authors showed that when mice were trained for 7 weeks, no significant differences exist between 3x-ET and 5x-ET groups. Is it possible to exclude a potential impact of training duration on such output? Is it feasible to assume that more frequent exercising per week would be more beneficial for patients when training would last for less than 7 weeks?

 Minor concerns:

1. Labeling of graphs in Figures 2,3,4, and 6 should be larger.

2. References have to meet MDPI requirements for reference style (formatting and font size).

Round 2

Reviewer 2 Report

After making the changes according to my suggestions, I am satisfied with this revised version. Therefore, I do not have any more comments for the authors.

Author Response

Dear Reviewer,

We would like to thank you for your time and careful consideration of our revised manuscript.

Your sincerely,

Maxime Pellegrin

Reviewer 3 Report

I am very satisfied with authors` response. All my concerns were addressed very well. The manuscript is now ready for publication.

Author Response

Dear Reviewer,

We would like to thank you for your time and careful consideration of our revised manuscript.

Yours sincerely,

Maxime Pellegrin